# Vitamin D as a Nutri-Epigenetic Factor in Autoimmunity—A Review of Current Research and Reports on Vitamin D Deficiency in Autoimmune Diseases

**DOI:** 10.3390/nu14204286

**Published:** 2022-10-14

**Authors:** Artur Mazur, Paulina Frączek, Jacek Tabarkiewicz

**Affiliations:** 1Institute of Medical Sciences, Medical College of Rzeszow University, University of Rzeszów, 35-310 Rzeszow, Poland; 2Department of Human Immunology, Institute of Medical Sciences, Medical College of Rzeszow University, University of Rzeszów, 35-310 Rzeszow, Poland; 3Centre for Innovative Research in Medical and Natural Sciences, Medical Faculty, University of Rzeszów, 35-310 Rzeszow, Poland

**Keywords:** nutrition, epigenetics, autoimmunity, vitamin D

## Abstract

Epigenetics is a series of alterations regulating gene expression without disrupting the DNA sequence of bases. These regulatory mechanisms can result in embryogenesis, cellular differentiation, X-chromosome inactivation, and DNA-protein interactions. The main epigenetic mechanisms considered to play a major role in both health and disease are DNA methylation, histone modifications, and profiling of non-coding RNA. When the fragile balance between these simultaneously occurring phenomena is disrupted, the risk of pathology increases. Thus, the factors that determine proper epigenetic modeling are defined and those with disruptive influence are sought. Several such factors with proven negative effects have already been described. Diet and nutritional substances have recently been one of the most interesting targets of exploration for epigenetic modeling in disease states, including autoimmunity. The preventive role of proper nutrition and maintaining sufficient vitamin D concentration in maternal blood during pregnancy, as well as in the early years of life, is emphasized. Opportunities are also being investigated for affecting the course of the disease by exploring nutriepigenetics. The authors aim to review the literature presenting vitamin D as one of the important nutrients potentially modeling the course of disease in selected autoimmune disorders.

## 1. Introduction to Epigenetic Modifications

The history of epigenetics is linked with the study of evolution and development. The first steps taken to understand epigenetics were based on observing the differentiation and specialization of cells and the adjustment of their metabolic functions with the knowledge that every somatic cell in the body contains the same genetic material. Since the beginning of interest in the topic of epigenetics, its definition has been changing and becoming more specific, leading to its current understanding as the presence of patterns of gene expression that can be inherited, and which are the foundation for changes in the phenotype. The pattern of these gene expressions is modified due to the influence of various factors, including the external environment (such as pollutants and radiation) and lifestyle (such as tobacco smoking, alcohol consumption, or inadequate diet). From studies on Agouti rodents and bee populations to studies involving a famine-exposed population during World War II, we have learned that food—the amount and type of nutrients—has an undeniable long-term effect on the lives of offspring, exerted by epigenetic mechanisms. As already mentioned, some of these mechanisms are hereditary, so epigenetic marks can potentially persist during development and be passed from offspring to offspring, which also affects the lives of grandchildren and possibly further generations [1,2].

Regulation of gene expression by activating or silencing genes is accomplished through simultaneous mechanisms that include i.a. DNA methylation, histone modification, and profiling of non-coding RNA. The process of DNA methylation is mediated by the DNA methyltransferase (DNMT) family, which transfers methyl groups on DNA strands. There are four known patterns of methylation, with the most widely studied being the methylation of CpG islands located in promoter regions of genes, and similar methylation of CpG island shores. Other patterns are methylation in gene bodies, and in repetitive sequences, which protects chromosomal integrity. Methylation results in silencing the gene. It can be done either by blocking the availability of heterochromatin for transcription in the promotor region or in the gene body itself. Among the DNMT family, we can distinguish subtypes, involved in methylation during embryonic development (de novo methylation), such as DNMT3A and DNMT3B, as well as those involved in methylation of hemimethylated sites generated during DNA replication, such as DNMT1. There is also a ten-eleven translocation (TET) enzyme family, described as having a role in the DNA active demethylation process. TET1-3 are dioxygenases catalyzing 5-methylcytosine oxidation and promoting cytosine demethylation. The TET enzymes group has been especially studied in stem cell differentiation and early development, and more recently in carcinogenesis in multiple solid cancers [3].

Histones are proteins responsible for packing and ordering DNA. The modification processes such as methylation, demethylation, acetylation, deacetylation, phosphorylation, ubiquitination, SUMOylation, and ADP-ribosylation reorganize the structure of chromatin, thus regulating gene expression. Acetylation is performed by histone acetyltransferases (HATs) and results in higher gene expression, while deacetylation is catalyzed by histone deacetylases (HDACs) and results in gene suppression predominance. These two best-known histone modifications occur on lysine residues in the N-terminal tail. It is worth noting that the region and the number of histone methylations may result in various effects—silencing or expression of genes. At the same time, the co-occurrence of DNA methylation and histone methylation can have different resultant effects. The methylation status of histones depends on dynamic processes determined by the activity of methyltransferases (HMTs) and demethylases (HDMs). An example of HDM can be amine oxidase lysine-specific demethylase 1 (LSD1/KDM1A), which was described to affect the clinical outcome and recurrence risk in various cancers, including colon cancer [4]. It is worth noting that depending on the histone substrate, H3K4me2/me1 or H3K9me2/me1, the result of the LSD1/KDM1A activity can be gene repression or activation, respectively [5].

Noncoding RNAs (ncRNAs) are sequences of infrastructural and regulatory functions with no protein product. Among regulatory ncRNAs, there are microRNAs (miRNAs), Piwi-interacting RNAs (piRNAs), small interfering RNAs (siRNAs), and long non-coding RNAs (lncRNAs), each with a slightly different regulatory function—modulating the transcriptional and post-transcriptional expression or repression of genes, regulating chromatin formation, histone modifications, and DNA methylation [6].

The impact of vitamin D has been noted in various epigenetic phenomena in recent years, mainly in the context of a search for the pathogenetic basis of cancer. In complex machinery of interactions, the genes responsible for proper vitamin D signaling may themselves undergo epigenetic modifications. Due to the long CpG islands in the promoter regions of genes essential for vitamin D-dependent signaling, these genes are susceptible to methylation and may be silenced. The interplay between the VDR gene product, its ligands, coactivators, corepressors, chromatin modifiers, remodelers, and the genes encoding these elements, creates a complex structure that is a likely substrate for the development of pathology. In terms of DNA methylation status, there was a negative correlation found between 1,25(OH)2D levels and CpG methylation in adenomatous polyposis coli—a tumor suppressor related to colon cancer [7]. In further studies, vitamin D seemed to alter colorectal cancer risk by mediating the Wnt regulatory genes—a strong negative association of vitamin D intake with DKK1 and Wnt5a methylation has been noted [8]. 1,25(OH)2D was also found to reduce DNA methylation of the e-cadherin promoter, or even promote DNA demethylation in human breast cancer [9,10]. The effect of vitamin D on the demethylation process was comprehensively described in the work of Fetahu et al. The authors gathered available data from studies describing the effect of VDR signaling on the expression of genes regulating oncogenic processes in patients with colorectal cancer and several other tumors. Attention was drawn to the cross-regulatory effect between VDR and histone demethylase activity. 1,25(OH)2D was found to increase the expression of the lysine-specific demethylase 1 and 2, JARID2, KDM5B, and KDM6B histone demethylases, while inhibiting others—KDM4A, KDM4C, KDM4D, KDM5A, KDM2B, JMJD5, JMJD6, and PLA2G4B. The effects of such broad actions are diverse, although they have been related, especially the inhibition of the KDM4 family, to genome stability, and the upregulation of KDM6B has been attributed to a possible antiproliferative role. [5,11] Higher vitamin D levels may also have an indirect influence on histone demethylases by mRNA [12,13].

A proper balance between epigenetic modifications is essential for embryonic development, division, differentiation, and maturation of all cells, including the cellular component of the immune system. As epigenetics has become a keenly explored topic in understanding the basis of cancer, its involvement is also being increasingly studied in the context of autoimmune diseases (ADs). The evidence so far has shown that epigenetic dysregulated modifications, including DNA methylation, histone modification, and ncRNAs, are involved in the pathogenesis of several ADs. Epigenetic mechanisms have also been discussed as a potential source of abnormal signaling interactions in various conditions, as well as the substrate of susceptibility of some individuals to developing different ADs [14,15,16].

## 2. Involvement of Vitamin D in Immunomodulation

Vitamin D in its most common form, cholecalciferol, is a micronutrient crucial for human health. Supplied from the diet or synthesized in the epidermis from 7-dehydrocholesterol, it has to be modified to obtain biological activity. Through progressive hydroxylation it is converted into 25-hydroxycholecalciferol (25(OH)D), and then the active form, 1,25-dihydroxycholecalciferol (1,25(OH)2D), a hormone that performs numerous functions. Interestingly, the conversion from 25(OH)D to 1,25(OH)2D occurs mainly in the kidneys with the CYP27B1, an enzyme stimulated by parathyroid hormone; however, the hydroxylation can be performed in other cells and tissues. Such an extrarenal hydroxylation, in various epithelial cells, parathyroid glands, and macrophages, is under the control of cytokines (e.g., TNFα or IFNγ) [17]. Circulating in the blood, 1,25(OH)2D binds with the specific vitamin D receptor (VDR), found in many tissues of the body, i.e., bone (osteoblasts and chondrocytes), muscle, adipose tissue, skin, epithelium, kidneys, pancreas, pituitary gland, as well as in immune cells: lymphocytes T, lymphocytes B, monocytes, natural killer cells, and antigen-presenting cells (APC) [18]. Apart from maintaining the calcium-phosphate balance and thus role in the correct structure and function of bones, vitamin D also takes a number of actions modulating immunological processes, which explains its interest in the context of autoimmunity. After binding to the VDR, vitamin D forms a heterodimer with the retinoid X receptor (RXR). The obtained complex activates the vitamin D response element and recruits enzymes responsible for histone acetylation. The ongoing structural changes in chromatin induced by the obtained complex result in targeted gene regulation [19]. More than 900 genes involved in both innate and adaptive immunity, as well as many other physiological interactions are under the control of 1,25(OH)2D and intracellular VDR connection. A deficiency of vitamin D has been therefore associated with a wide range of diseases, including cardiovascular, neuromuscular, and metabolic disorders, hypertension, neoplasm, as well as infectious and autoimmune diseases [20,21]. In terms of the innate response, vitamin D is proven to activate NK cells and NKT cells modulating the secretion of cytokines, such as IFN-γ and IL-4, with simultaneous suppression of macrophage activation and the domination of the M2 “anti-inflammatory” macrophages. On the other hand, the antimicrobial activity of macrophages induced by vitamin D is known—both the increased phagocytosis of *M. tuberculosis* and *P. aeruginosa*, as well as the induction of cathelicidin and defensin secretion [22,23,24]. The protective function of 1,25(OH)2D through genomic and non-genomic mechanisms has been recently studied in COVID-19 patients. Worth noting is the non-genomic activity of vitamin D against SARS-CoV-2, involving active inhibition of the replication machinery of the virus. According to Qayyum et al., vitamin D and the inhibitory action of luminesterol on the M^pro^ viral protease and the viral RNA-dependent RNA polymerase may play a significant role in an active fight against SARS-CoV-2 infection and thus diminish the severity of COVID-19 progression in patients [25,26]. The infectious stimulus and the type of cell-producing cytokine seem to be crucial for the impact of vitamin D on the production of IL-8. 1,25(OH)2D may increase IL-8 production and thus the ability of neutrophils to respond to invading pathogens by recruiting additional neutrophils to the site of infection. However, vitamin D has been also described to decrease IL-8 release in hyperinflammatory macrophages [27]. Vitamin D is known to inhibit monocyte production of proinflammatory cytokines such as IL-1, IL-6, IL-12, and TNF-α. It also restrains the differentiation and maturation of dendritic cells with decreased expression of MHC class II molecules, co-stimulatory molecules, and IL-12, important in self-tolerance [28]. Studies show that dendritic cells can metabolize vitamin D for the programming of T cells. What is more, 1,25(OH)2D may also interact with dendritic cells directly and influence their migration and capacity to instruct T cells and hence to initiate, fine-tune, or suppress immune responses [29]. The role in promoting proliferation and effector functions of immunosuppressive T regulatory cells is especially explored. Vitamin D supplementation may increase Treg/CD3 ratios in both healthy individuals and patients with autoimmune disorders as well as the T regulatory cells function [30]. Vitamin D is responsible for the activation and proliferation of lymphocytes, and the differentiation of Th lymphocytes, resulting in a more balanced Th1/Th2 response that limits the development of self-reactive T cells preventing inflammation and autoimmunity [31]. This includes regulation of the Th17 lymphocyte population through an alternative pathway of 20-hydroxy- and 20,23-dihydroxyvitamin D synthesis involving related orphan receptors α and γ (RORα and RORγ). This is possible as RORα and RORγ have also been identified as major regulators of the said lymphocytes [32]. Moreover, the differentiation of B lymphocytes into plasma cells and the production of antibodies is under the inhibitory influence of vitamin D. It should be remembered that, although naïve B cells have a relatively small expression of VDR mRNA, the authors of the present study observed a three- and fourfold increase in VDR expression in the excited state with the use of anti-CD40/IL-21 and anti-IgM/anti-CD40/IL-21 [33]. In addition, vitamin D-mediated signaling inhibits the formation of memory B cells as well as immunoglobulin secretion in activated B cells. The IgA, IgG, and IgM production is suppressed, while data concerning IgE secretion are conflicting, which may be accounted for by the modulatory role of vitamin D and IL-4-dependent increase in IgE immunoglobulins due to Th2 skewing [34]. Broad immunomodulatory effects on cell populations of the immune system and their functional changes were presented by Cyprian et al., and earlier by Mora et al. [35,36]. The mechanisms described above are summarized in Figure 1.

Maintaining proper vitamin D levels is therefore crucial for physiological immune functions. At the same time, according to some sources, vitamin D deficiency is nowadays considered the most common medical condition, affecting more than a billion people worldwide [37]. A 25OH2D of <50 nmol/L or 20 ng/mL is defined as vitamin D deficiency. It affects the European population in varying degrees—from <20% of the population in Northern Europe, 30–60% in Western, Southern, and Eastern Europe (even around 90% in Poland) [38], and up to 80% in the Middle East. Similar trends with the range of 20–90% have been reported for Australia, India, Africa, South America, Turkey, and Lebanon, suggesting that vitamin D deficiency is a problem for both developing and developed countries [39]. Severe deficiency (serum 25OHD < 30 nmol/L or 12 ng/mL) is found in >10% of Europeans and in >20% of the populations of India, Tunisia, Pakistan, and Afghanistan [40]. These figures may vary by ethnicity in different regions of the countries listed, as well as by age groups, with lower vitamin D levels occurring in children and the elderly [37,41].

Knowing that the vitamin D status of the fetus and the newborn is completely dependent on the vitamin D levels in maternal blood, we better understand the importance of adequate supplementation during pregnancy and lactation in terms of proper modulation of the child’s immune responses [42]. Studies in various countries have shown that vitamin D deficiency in pregnant women and infants is common, affecting 4% to 60% of the former and 3% to 86% of the latter [43,44,45]. In fact, adequate vitamin D levels determine the normal course of pregnancy from a very early stage, due to the necessary tolerance of the semi-allogenic fetus by the mother’s immune system. Observational studies have linked vitamin D deficiency with preeclampsia, altered placental vascular pathology, abnormal fetal growth patterns, as well as the risk of preterm delivery [46].

In children, vitamin D deficiency is classically associated with the occurrence of rickets. In recent decades this disease remains a significant public health disorder despite the fortification of food. This is mainly due to the prevalence of the described deficiency. While skeletal symptoms are the most recognizable, it is the extraskeletal complications, hypocalcemic seizures, and cardiomyopathy that are the most devastating and cause the reported fatalities [47]. Newborns with vitamin D deficiency usually do not have overt defects in skeleton or calcium metabolism, while deficiency has been linked to a higher risk of a number of disorders in this age group: respiratory distress syndrome, lower respiratory tract infections, food sensitivities, asthma, autism, schizophrenia, and type I diabetes [42,48,49,50].

## 3. Vitamin D Deficiency and Epigenetic Dysregulations in Autoimmunity

In autoimmune conditions, the immune system misidentifies the host’s own cells and tissues as foreign elements, which results in developing an immune response against them. According to a commonly accepted definition, in the background of ADs are certain genetic predispositions of an individual, which are superimposed by external factors—the so-called triggers that cause the disease to manifest. Undoubtedly, external factors can affect the epigenetic mechanisms described earlier and thus contribute to the development of ADs. Numerous interactions between epigenetic modifications and ADs, both systemic and organ-specific, have been identified. These correlations have been described in rheumatoid arthritis (RA), systemic lupus erythematosus (SLE), systemic sclerosis (SSc), Sjögren’s syndrome (SjS), inflammatory bowel disease (IBD), multiple sclerosis (MS), type 1 diabetes (T1D), and others [16]. Reports of low serum vitamin D predicting the course of ADs in the future have been published [28]. Vitamin D has also been shown to facilitate the progression of existing autoimmune diseases. In the study with undifferentiated connective tissue disease (UCTD) patients, the mean vitamin D level was significantly lower in the group that progressed to a definitive disease [51]. Disease activity has also been shown to correlate inversely with vitamin D in many but not all studies.

### 3.1. Systemic Autoimmune Diseases

SLE is believed to be the most studied autoimmune disorder correlated with epigenetic modifications. Among patients with ADs, a higher prevalence of vitamin D deficiency was commonly observed in SLE patients, which may be due to increased photosensitivity or possible renal complications of the disease, disrupting the effective hydroxylation of 25OHD [28,35]. At the same time, lower 25OHD levels found in SLE patients suggest that vitamin D deficiency may be a risk factor for the disease. The studies have also found higher SLE disease activity measured with the Systemic Lupus Erythematosus Disease Activity Index (SLEDAI) associated with lower levels of vitamin D [52]. Similar correlations between low levels of vitamin D and disease activity and severity have been observed in other ADs such as MS and RA [28]. In their broad review, Mazzone et al. presented the importance of epigenetic alterations in ADs, including SLE patients. Attention was drawn to the methylation status of specific genes, such as CD11a (ITGAL), perforin (PRF1), CD70 (TNFSF7), and CD40LG (TNFSF5) in T lymphocytes as a factor in SLE pathogenesis and development. Histone modification patterns and the role of ncRNAs in SLE have been investigated to a lesser extent than DNA methylation. In general, the hypomethylation also observed in SLE is a result of the restricted availability of methyl donors and/or disruption of DNMT1 activity. Therefore, a balance between methyl donor, S-adenosylmethionine (SAM) as a factor dependent on dietary micronutrients, such as folate, zinc, methionine, choline, and vitamins, and DNMT1 seems to be the key to avoiding SLE flares [53]. A recent meta-analysis by Yang et al. evaluated the effect of VDR gene polymorphisms on susceptibility to developing SLE. As the researchers point out, links have been demonstrated between polymorphisms and different populations: between the VDR ApaI polymorphism and the susceptibility of the general population, between the VDR BsmI polymorphism and the susceptibility of the African and Caucasian populations, and between the VDR FokI polymorphism and the susceptibility of the African population [54]. The aforementioned polymorphisms—ApaI, BsmI, and FokI—are among the four polymorphic sites within the VDR gene, in addition to the TaqI polymorphism. The ApaI and BsmI sites are located in the intron portion of the gene, FokI in exon-2 is responsible for the creation of an alternative transcription initiation site, while TaqI in exon-9 is the result of a silent T to C substitution. All the mentioned sites can disrupt vitamin D metabolic pathways [55]. Although referenced study detected no significant correlation between TaqI polymorphisms and susceptibility to SLE in the populations assessed, further explorations concerning this conclusion need to be conducted. Dietary interventions in SLE are a frequently addressed topic of research since in this group of patients it was quite common to find vitamin and mineral deficiencies. Proper supplementation provides antioxidant, anti-inflammatory, and immunomodulatory effects, and thus makes it possible to reduce the severity or prevent the disease [53]. Currently, a diet rich in vitamins and minerals, and mono- and polyunsaturated fatty acids with moderate energy consumption preventing obesity and potentially cardiovascular disease is recommended to control the inflammatory findings of the disease and the complications and co-morbidities resulting from SLE therapy [56].

Although the incidence of RA does not begin to increase until after the age of 25, due to interesting reports of correlations between single nucleotide polymorphisms (SNPs) in vitamin D metabolic pathway genes and susceptibility to RA, it was decided to include it in this review. Multiple epigenetic factors in the pathogenesis and progression of the disease have already been identified. They include the role of RA synovial fibroblasts (RASFs), which through epigenetic modifications become a source of numerous pro-inflammatory cytokines. A participatory global hypomethylation in RASFs, associated with the acute course of the disease, has also been described before [57]. In a study by Tian-Ping Zhang et al., ten SNPs in vitamin D metabolic pathway genes (CYP2R1, CYP24A1, VDR, and CYP27B1) were genotyped in an RA patient and a control group. The results showed that CYP2R1 and CYP27B1 genetic variations were associated with the genetic background of RA, while altered VDR and CYP27B1 methylation levels were related to the risk of RA [58]. The study also confirmed that vitamin D deficiency is prevalent among RA patients, and the 25OHD level is significantly lower compared with healthy controls. Subsequent studies yielded similar findings in this regard. At the same time, in view of previous reports of an increased risk of RA among Caucasians due to VDR gene polymorphisms and the FokI variant, an analysis of the association of four selected VDR gene polymorphisms (BsmI, FokI, ApaI, and TaqI) with susceptibility to RA in the Lithuanian population was conducted. However, the genetic analysis did not reveal RA susceptibility in the study group. Instead, a significant inverse correlation between vitamin D levels, DAS28, CRP, and HAQ scores was noted, indicating an association of vitamin D deficiency with increased disease activity and disability scores in these patients [59].

In SjS, the influence of vitamin D levels is still controversial. On the one hand, the lack of UV exposure as part of the treatment of cutaneous manifestations of the disease has been postulated as a risk factor for vitamin D deficiency, while on the other hand, the clinical picture of the disease and at least some of its manifestations appear to be reliant on vitamin D deficiency [60]. The deficiency has been linked to the appearance of peripheral neuropathy and non-Hodgkin lymphoma (NHL) in patients with SjS, which translates into mortality in this AD [61]. In terms of the impact on the nervous system, vitamin D was described to participate in the biosynthesis of neurotropic factors, production of enzymes for neurotransmitter synthesis, inhibition of inducible nitric oxide synthase (iNOS) synthesis as well as increasing levels of glutathione and gamma-glutamyl-transpeptidase [60,62]. Therefore, a correlation is being sought between vitamin D deficiency in patients with SjS and the severity of sensory or motor-sensory neuropathy, as well as abnormal corneal innervation and potentially more severe discomfort on the ocular surface [63]. Ro/SSA and La/SSB antibodies are the key serological findings associated with a congenital cardiac block in neonates of mothers with SjS. The vitamin D status was also studied as a potential factor for cardiac block development in children of SjS patients. Based on the observation of seasonal influence on the development of anti-Ro and anti-La positive congenital cardiac block, Ambrosi et al. suggested that low levels of vitamin D during 18 to 24 weeks of pregnancy could be involved in the pathogenesis of this disease. They found that average vitamin D levels for each month of pregnancy inversely correlated with the proportion of congenital heart block pregnancies [60,64,65]. Epigenetic studies are still needed to describe the complexity of primary SjS. As outlined in a recent review analysis by Imgenberg-Kreuz J, within the HLA genes we find the strongest genetic association with the occurrence of pSS. Attention is also drawn to the role of DNA hypomethylation and interferon-induced gene overexpression [66]. An association was also sought between the occurrence of VDR polymorphisms and susceptibility to primary SjS. In the currently only study on a Hungarian group evaluating this phenomenon, no such association was found for the BsmI, ApaI, TaqI, and FokI polymorphisms [67].

In patients with SSc—a chronic disease with vasculopathy, and visceral and cutaneous fibrosis—vitamin D deficiency of ≤30 ng/mL has been reported in up to more than 87% of cases [68]. Poor vitamin status seems to be related to a more aggressive disease with multi-visceral and severe organ involvement—especially pulmonary and cardiac involvement. When comparing 25OHD levels, Groseanu et al. found no difference between diffuse and limited subtypes of the SSc [69]. In the systemic literature review by Schneider et al. the authors conclude that despite the vast literature on vitamin D deficiency and SSc incidence, it remains unclear if hypovitaminosis is an epiphenomenon or if it actually determines an increase in susceptibility and a worse prognosis for this complex disease. Analyses to date have postulated the involvement of impaired VDR signaling with reduced expression in fibroblasts from SSc patients and overactivation of TGF-β signaling. The study by Juan Li et al. selected as many as eight single nucleotide polymorphisms (TaqI, FokI, ApaI, BsmI, Cdx2, BglI, Tru9I, and rs11168267) of the VDR gene and evaluated the association between their occurrence and susceptibility to SSc. It has been shown that ApaI and BglI genotypes may be important in the pathogenesis of SSc, while no significant association was found for the other single nucleotide polymorphisms. At the same time, no association was observed between the polymorphisms studied and the clinical picture of SSc [70]. Thorough research on the clinical effects of vitamin D supplementation in SSc patients as well as the clarification of vitamin D commitment in SSc pathogenesis is still needed [71].

The vitamin D status and the VDR function have also been of interest in studies on IBD. The expression level of VDR is high in the intestine, and therefore the role of surveillance of cell proliferation, barrier function, and immunity has been attributed to it. Vitamin D deficiency, low VDR expression, and dysfunction of vitamin D/VDR signaling have been observed in patients with Crohn’s disease (CD) and ulcerative colitis (UC), and they were found to be related to the activity in both diseases [72]. Studies have shown that a low level of intestinal epithelial VDR is accompanied by a reduction in Atg16l1, an IBD risk gene, and a regulator of autophagy, which leads to dysbiosis and impaired autophagic responses [73]. Moreover, impaired vitamin D/VDR signaling fails to regulate the proper expression of several components of tight junctions and adherent junctions, which causes disruption of the mechanical barrier of the intestine and leads to a less effective mucosal healing process in murine models [74]. In a systematic electronic search by Vernia et al., the authors stated that vitamin D deficiency in IBD is multifactorial, resulting from inadequate sun exposure, unnecessary dietary restrictions, and, in some instances, impaired absorption of nutrients—a possible effect of an ongoing inflammation. Emerging evidence suggests that vitamin D deficiency may be implicated in a more aggressive disease behavior and an impaired response to biological therapy [75]. Deficiency of 25OHD has been associated with more hospitalizations, flare-ups, use of steroids, and escalating treatment [76]. Therefore, as vitamin D supplementation may prove to be one of the important elements of therapeutic management in IBD patients, reliable evidence is being sought to determine the dose of vitamin D effective for intervention. A limitation of most of the available data attempting to elucidate the molecular mechanisms of vitamin D action, including the effect on the maintenance of the mucosal barrier in the intestinal lumen, is that it comes from studies conducted in preclinical settings, making it difficult to translate the results into clinical management. The intestinal microbiota, which is a component of the vitamin D–IBD axis, should not be forgotten either. The ratio of commensal to pathogenic bacteria and its effect on the VDR was one of the elements studied within this topic [77,78]. The interplay of bacterial metabolites, butyrate, and lithocholic acid, as well as bacterial enzymes (e.g., capable of activating vitamin D) and growth factors on VDR signaling and the course of inflammation, has also been explored [75]. Guidelines on the practical use of probiotics as valuable components of treatment in IBD are also expected in the future [79].

### 3.2. Organ-Specific Autoimmune Diseases

The increasing incidence of type I diabetes mellitus (T1DM) has led to an active search for information on the interplay between diet and epigenetics in this disease. Understanding the diet–epigenome axis with vitamin D as one of the nutritional factors involved could potentially allow new diagnostic and therapeutic approaches for patients with T1DM. The pathogenesis of T1DM highlights known environmental factors associated with a higher risk of its development, including infections, dietary factors, advanced maternal age, psychological stress, antibiotic use, mode of delivery, and steroid intake [80]. Viral infections—especially by group B coxsackieviruses or echoviruses—have also been long considered as one of the most likely trigger candidates for T1DM [81]. Diet and nutrients as potential triggers of T1DM are still being studied with some interesting yet inconclusive insights. Evaluated dietary factors involve breastfeeding, early intake of cows’ milk, solid foods (fruit, root vegetables, gluten, and non-gluten-containing cereals, and eggs), and vitamin D [82]. Most of the cohort studies showed that breastfeeding and vitamin D had protective effects, whereas bovine milk and the early introduction of gluten were risk factors [83]. As Cerna et al. noted, these risk factors support the hypothesis that general antigenic stimulations are more important than actual antigens in the disease process. Combining these compounds with an immature immune response and insufficient tolerance in the gut, as well as a predisposition to inflammation due to a deficiency of long-chain polyunsaturated fatty acids, typical of the western diet, the hypothesis seems credible [84]. The concept of the influence of the gut microbiota on increasing or decreasing the risk of developing autoimmune diseases, including T1DM, is consistent with the above observations. Although the topic of the effect of breast milk on T1DM is controversial and one can also find sources in the literature that have not shown its protective role in the disease, it is believed that breastfeeding reduces intestinal epithelial permeability by preventing triggering factors [85,86]. In the past, vitamin D deficiency has been linked to the occurrence of T1DM, among others, based on epidemiological data showing that countries at northern latitudes had a high prevalence of T1DM [83]. Compared to healthy individuals, patients with T1DM also had lower 25OHD values [87,88,89,90]. This correlation was also supported by reports that vitamin D supplementation lowers the risk of developing T1DM, and that when supplemented appropriately (more than 30 ng/mL), vitamin D can help preserve residual ß-cell and insulin secretion, as well as improve glycemic control and insulin sensitivity [91,92]. A historical large-scale birth cohort study in Finland evaluated the effect of vitamin D supplementation on rickets and the development of type 1 diabetes and found an 80% reduction in the risk of T1DM in children who received >2000 IU of vitamin D per day compared to children receiving less or no vitamin D supplementation [93]. However, in a study by Bierschenk et al. comparing serum levels of 25OHD in patients with type 1 diabetes, first-degree relatives, and controls, the level of 25OHD was found to be low in all groups, and not specifically associated with T1D. The authors concluded that the uniform suboptimal 25OHD levels, despite residence in a zone with abundant sunshine, support additional dietary vitamin D fortification practices [94]. Important conclusions in terms of prenatal prevention of T1DM were reached in their study by Marjamäki et al. They found that maternal intake of vitamin D from food or supplements during pregnancy was not associated with advanced beta cell autoimmunity/type 1 diabetes or type 1 diabetes itself in Finnish offspring having increased genetic susceptibility to T1D [95].

Attention is also drawn to the role of the VDR and its polymorphism in T1DM. In recent years, a number of studies have examined the association of VDR gene polymorphisms with T1DM risk in different populations, with conflicting results. A recent meta-analysis of 39 case-control studies by Zhai et al. rejected any significant association between VDR gene polymorphisms and T1DM risk in the overall results. At the same time, the results of subgroup analysis revealed significant negative and positive associations between FokI and BsmI polymorphisms and T1DM in Africans and Americans, respectively. The authors emphasize that compared to the previous meta-analysis from 2014 by Tizaouia et al., apart from the association of VDR genetic polymorphisms with T1DM risk in different ethnic groups, the overall analysis was almost the same despite including further studies [96,97]. It is not entirely clear where all the differences in the observed vitamin D–diabetes axis of interactions come from. As Kohil et al. explain, some of these discrepancies can be attributed to differences in the type of supplement used (i.e., cholecalciferol, alpha-calcidiol, or calcitriol), the dose of the vitamin, the age group of the study participants, and/or the duration of diabetes [80]. Apart from the VDR polymorphism, the authors highlighted the potential role of CYP27B1 polymorphisms with a favorable genetic background for various autoimmune disorders including T1DM [82]. Knowing that the immune cells are also expressing CYP27B1 and thus modulate the immune response, the studies previously performed on the European population (German, British, and Polish) provide significant data concerning a potential decrease in the availability of the active form of vitamin D in people with the CYP27B1 promoter C(-1260)A polymorphism [98,99,100,101].

The pathogenesis of autoimmune thyroid diseases (AITDs) has been suggested to be multifactorial, including genetic, environmental, and hormonal factors, such as vitamin D deficiency [102]. A recent review by Lee et al. focused on understanding the role of immune-related genes and thyroid-specific genes gathered in the group of AITD susceptibility genes. Although the topic of vitamin D metabolism and the genes and epigenetic mechanisms affecting it was not directly addressed in the paper, the undoubted involvement of IFN-α in triggering thyroid autoimmunity was highlighted [103]. Due to the conflicting results of previous studies on VDR polymorphism and the incidence of AITDs, a meta-analysis of eight studies with a total number of participants exceeding 1000 with AITDs was conducted. Four of the most commonly described polymorphisms of the VDR gene—BsmI, FokI, ApaI, and TaqI—were evaluated, showing that the BsmI or TaqI polymorphism is significantly associated with AITD risk. However, the authors noted important limitations of the analysis related to the omission of gene-environment interactions, the lack of a uniform definition of the control group, and the limited number of studies and their participants, especially in the context of patient ethnic diversity [104]. Vitamin D deficiency is highly prevalent in endocrine disorders and its supplementation appears to have beneficial effects, as described in one of the recent reviews by Galușca et al. In an analysis of the literature on Hashimoto’s disease, low vitamin D production appeared to be associated with higher anti-thyroid peroxidase (anti-TPO) antibody titers and thyroid volume, while supplementation was addressed in connection with a reduction in antibodies levels in some studies. In addition, some researchers indicated a gradual decrease in thyroid-stimulating hormone (TSH) levels with supplementation. The serum concentration of vitamin D showed a significantly lower value in Graves’ disease patients who were not in remission compared to those who were. Moreover, in a randomized prospective study, the thyroid volume and the degree of exophthalmos revealed a statistically significant correlation with vitamin D levels [105,106].

MS is an inflammatory disease characterized by neurodegenerative events and autoimmune attacks on myelin in the central nervous system (CNS), leading to varying degrees of recurrent or progressive neurological disorders. Epigenetic changes play an important role both in the development of myelination and remyelination and in the pathogenesis of some neurodegenerative diseases, including MS, Alzheimer’s, Parkinson’s, and Huntington’s diseases [107]. Relapses of MS usually occur in the absence of a defined trigger; however, it has been described that MS symptoms vary throughout the year, suggesting that environmental factors may act as the above-mentioned triggers, and influence the overall susceptibility to MS and its progression [108]. Vitamin D is one of the best-described environmental factors for MS. It has been found to modulate Th17 autoimmunity through transcriptional suppression of the pro-inflammatory cytokine IL-17, via recruitment of histone deacetylase 2 to the *IL17A* promoter region. Moreover, 1,25(OH)2D may change the expression of genes that modify histones [107,108]. In a comprehensive review by Sintzel et al., the authors state that there is increasing evidence indicating a causal relationship between low vitamin D levels and the risk of MS as well as greater clinical activity and brain MRI-confirmed activity in MS patients [109]. Interventional studies conducted so far on limited groups of MS patients have shown that a high supplemental dose of vitamin D (10,400 IU/day) is safe and exhibits pleiotropic immunomodulatory effects. A reduction was found in the proportion of interleukin-17+CD4+ T cells, CD161+CD4+ T cells, and effector memory CD4+ T cells with a concomitant increase in the proportion of central memory CD4+ T cells and naive CD4+ T cells. These effects were not observed in the group of low-dose supplementation (800 IU/day) [110]. More research is needed to confirm whether high-dose supplementation of vitamin D in MS patients is truly beneficial in the context of changes in calcium levels and prolonging the time to disease progression to the secondary progressive phase. The search for a link between VDR polymorphisms and MS is still a topic under investigation. However, information on this relationship is heterogeneous, likely influenced by matched research groups. In a recent retrospective study involving more than 200 patients with relapsing-remitting multiple sclerosis and more than 800 Caucasian healthy controls, no influence of the ApaI, BsmI, Cdx2, and TaqI was found. However, a significant effect of the VDR Fokl (rs2228570) on the development of MS was demonstrated [111]. It should be noted, however, that in an earlier large meta-analysis, despite notable inconsistencies between the reports evaluated, a significant association between TaqI polymorphism and MS susceptibility was detected, while the effect of the BsmI polymorphism on increasing the risk of MS was detected only in the Asian population. Interestingly, it was also found that the ApaI VDR polymorphism may have a protective function, reducing MS risk in the Asian population [112]. The process of DNA methylation should also be mentioned as a regulator of processes in MS pathogenesis. In addition to the hypomethylation of FOXP3 or IL-17 in T cells, the methylation pattern of the alternative VDR promoter, which is located in exon 1c, has also been studied. This gene regulatory element was previously found to be hypomethylated in other immune cells; however, in a study by Ayuso et al., a different, moderate pattern of methylation in the T cells population was found. What is more interesting, in the group of relapsing-remitting multiple sclerosis patients the methylation of the exon 1c promoter was increased even more. The search for the significance of this VDR promoter methylation pattern in MS patients, as well as in patients with other autoimmune diseases, is a challenge for further research [113].

## 4. Current Clinical Trials Using Vitamin D in Autoimmune Diseases

Literature sources to date point to the need for vitamin D supplementation to compensate for vitamin D deficiency, which should be pursued in all individuals, not just those with autoimmune disorders. However, even the guidelines for the overall vitamin D supplementation dose in the community are not standardized. While in the United Kingdom, guidelines indicate 400 IU per day as the recommended dose for adults and children over the age of 4, the Endocrine Society recommends a daily supplementation dose of 600–1000 IU depending on age, and in cases of deficiency, 2000 IU per day, even to 50,000 IU per week. It seems that prevention and treatment of different conditions require different concentrations of vitamin D. Cutoff values in vitamin D concentration supported by high-level evidence studies are still not available. However, it appears that for interventions in bone mineral density, dental health, fracture risk, or tumorigenesis, benefits are obtained starting at 25OHD concentrations of 30 ng/mL [114]. For tuberculosis prevention, in vitro studies have indicated a cholecalciferol concentration of 4 µg/mL as adequate to slow proliferation in culture. This is a much higher concentration, but potentially achievable due to local hydroxylation performed by macrophages. At the same time, in vivo studies did not provide a clear conclusion [114,115]. A number of research and review papers, including those cited in this paper, point to the need for robust randomized controlled trials on large groups of patients to determine a dose of vitamin D supplementation that would demonstrate long-term benefits with a safety profile in ADs. In their study, Fletcher et al. point out other important uncertainties regarding the supplementation dose used. Indeed, it is still unclear what serum 25OHD level is optimal for immune function. It is possible that different levels of 25OHD are optimal for innate antimicrobial and antiviral responses relative to anti-inflammatory effects. It is also possible that the level of 25OHD beneficial in AD differs from that considered appropriate for maintaining skeletal health [116]. Prominent studies in the search for supplementation doses in people with ADs are those evaluating intervention with vitamin D in patients with MS. Notable are the exceptionally high daily doses of vitamin D (from 10,000 to 40,000 IU/day) used in these clinical trials, which has proven safe as a combined 25OHD–interferon-beta therapy. Only a few of the studies (presumably on a sufficient study group) have shown promising results in terms of improved MRI results, although baseline results were not achieved [117]. The literature also emphasizes the importance of the risk of vitamin D toxicity, especially in the face of hazardous internet trends, encouraging MS patients to use ultra-high doses of vitamin D of 80,000–100,000 IU daily [109]. For other ADs, positive effects of vitamin D are also sought, but lower supplementation doses are usually considered. A recent interventional study in patients with SLE showed benefits in disease activity and fatigue after 25OHD supplementation as follows: 8000 IU daily for 4 weeks, followed by 2000 IU daily maintenance for patients with vitamin D insufficiency, or 8000 IU daily for 8 weeks, followed by 2000 IU daily maintenance in vitamin D-deficient patients [118]. Another missing piece is solid evidence for vitamin D use in autoimmunity prevention. As a model on this subject, a recent study with 25,871 participants, supplemented with a placebo, omega-3 fatty acids, or vitamin D (2000 IU/day), should be mentioned here. Vitamin D supplementation, with or without omega-3 fatty acids, was shown to reduce the incidence of autoimmune diseases in this cohort by 39% after 3 years of follow-up and by 22% after 5 years [119]. It is therefore suggested that vitamin D use may be considered as prevention against ADs, but may also require long periods of supplementation. Despite the availability of studies based on animal models, as well as numerous studies of vitamin D and specific autoimmune diseases in humans, high-quality randomized trials should be pursued in this area as well [116].

Listed below are some interesting clinical trials using vitamin D in patients with ADs. All were selected based on a search of clinicaltrials.gov and include registered, currently ongoing studies. They may inspire further research efforts and be the basis for new dietary recommendations in the management of autoimmune diseases in the future.

“Pilot Study of OMEGA-3 and Vitamin D in High-Dose in Type I Diabetic Patients (POSEIDON)”—NCT03406897. A recruiting interventional open-label study with an estimated 56 participants. The authors aim to evaluate the efficacy and safety of a treatment regimen based on omega-3 fatty acids and vitamin D in patients with T1D. The authors hypothesize that the used combination administered to patients with new or established forms of the disease will be safe and preserve insulin secretion [120];“Early High-Dose Vitamin D and Residual β-Cell Function in Pediatric Type 1 Diabetes”—NCT05270343. Not yet recruiting interventional open-label study with an estimated 198 participants. The aim of the project is to evaluate the effect of high-dose vitamin D supplementation on early T1D in children and adolescents. Patients simultaneously require intensive insulin therapy [121];“Effect of Vitamin D Supplement on Disease Activity in SLE”—NCT05260255. The purpose of the ongoing study is to evaluate the effect of vitamin D supplementation on SLE activity (Systemic Lupus Erythematosus Disease Activity Index 2000). A recruiting interventional double-blind study with an estimated 100 participants. The study is also expected to assess IL-6 levels and anti-dsDNA titers at specific intervention intervals [122];“The Vitamin D in Pediatric Crohn’s Disease (ViDiPeC-2) (ViDiPeC-2)”—NCT03999580. The aim of the study is to evaluate the effectiveness of high-dose vitamin D supplementation in children with CD. The researchers expect to achieve a reduction in the frequency of relapses and improved patient quality of life [123];“High Dose Interval Vitamin D Supplementation in Patients with Inflammatory Bowel Disease Receiving Biologic Therapy”—NCT04331639. A recruiting interventional open-label study with an estimated 50 participants. Vitamin D will be administered concurrently with IBD biologic therapy every 4–8 weeks. The researchers aim to evaluate, using laboratory tests and a questionnaire, inflammatory markers as well as markers of bone health after intervention with vitamin D [124];“Longitudinal Effect of Vitamin D3 Replacement on Cognitive Performance and MRI Markers in Multiple Sclerosis Patients”—NCT03610139. A recruiting interventional single-blind study with an estimated 162 participants. The researchers hypothesize that high-dose vitamin D supplementation will result in improvements in cognitive performance at 6 and 12 months after supplementation [125].

## 5. Conclusions

The above literature review represents a collection of studies and observations to date on the epigenetic function of vitamin D in autoimmune diseases. The latest available reports exploring the epigenetic basis of selected disease entities and the involvement of vitamin D receptor relay in the most commonly considered epigenetic mechanisms, as well as the possible benefits of vitamin D supplementation in patients, are analyzed. An effort was made to present the sometimes contradictory reports in an accessible manner. Using the clinicaltrials.gov database, the ongoing interventional clinical trials evaluating the utility of vitamin D in ameliorating the course of autoimmune diseases and improving quality of life were presented. Further challenges in addressing the broad topic of vitamin D in autoimmunity were also identified.

## Figures and Tables

**Figure 1 nutrients-14-04286-f001:**
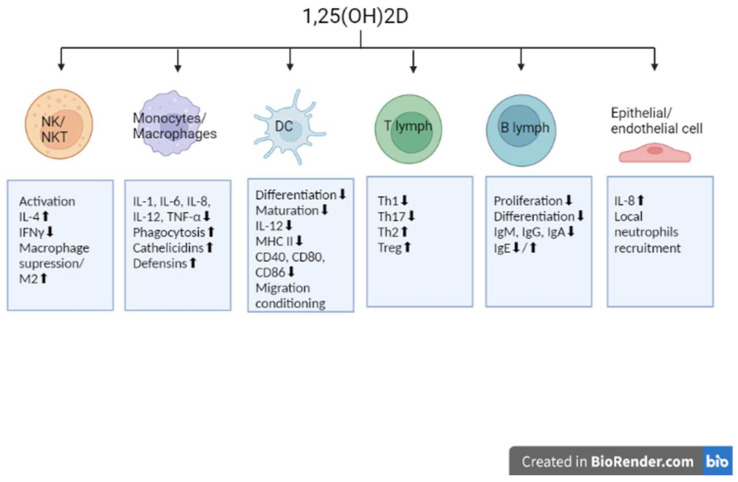
Immunomodulatory effect of 1,25(OH)2D on selected cells. NK, natural killer; NKT, natural killer T; DC, dendritic cell; T lymph, T lymphocyte; B lymph, B lymphocyte. Created with BioRender.com.

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
