# Peer review of "Vitamin D as a Nutri-Epigenetic Factor in Autoimmunity—A Review of Current Research and Reports on Vitamin D Deficiency in Autoimmune Diseases"

_nutrients, 2022, doi:10.3390/nu14204286_

Round 1

Reviewer 1 Report

This review is described about the relation between vitamin D and autoimmune disease, including epigenetic information. Overall, this is interesting and well-written.

This paper is interesting but it would be more interesting if the relationship between epigenetics and vitamin D were described in more detail for each disease. 

Author Response

Thank you very much for your favorable opinion. As recommended, the section on specific epigenetic processes that are known to model vitamin D - VDR receptor signaling has been expanded. Corresponding information has been added along with the source material so that relevant data appears in each disease entity listed. We hope that we have met the requirements and with the other changes suggested by the other reviewers, the paper will receive a positive review.

Reviewer 2 Report

The work presented is characterized by a review, it is interesting and current, but does not bring any major update on the subject. the issue of vitamin D and its epigenetic action is already well discussed in the literature. Perhaps the greatest contribution is the analysis of contradictory data on the action on the immune response, mainly with regard to the specific organ question of the response to vitamin D. I believe that some figures of the mechanisms of action would make the work more interesting

Author Response

Thank you very much for your comment. The topic of vitamin D and its multifaceted effects has indeed been the subject of research for years, however, it is worth noting that in this seemingly well-known topic, new analyses are still emerging in an attempt to support specific clinical recommendations. We believe that one such still-explored direction is the issue of vitamin D, epigenetics (or more precisely, nutriepigenetics) and autoimmune diseases, hence the inspiration for the manuscript in question. As recommended, a figure has been included to summarize the paragraph on the mechanisms of action of vitamin D on selected cells described in the text. We hope that in its current form the topic is clearer and more accessible. At the same time, with other modifications suggested by other reviewers, we hope that the work will receive a positive review.

Reviewer 3 Report

Summary

The authors present a comprehensive update of vitamin D control of the immune system in autoimmune disorders and epigenetic alterations contributing to the abnormalities.

There are some important clarification and corrections to be implemented:

1.       Lane 146-148: “Small amount of VDR in the cell surface of immune cells”. Please correct. The authors indicate at the beginning of the article that the VDR is an intracellular receptor.

2.       Please, indicate in the introductory paragraph on epigenetics, the main reported roles of vitamin D in the control of demethylases, as described in well cited articles such as Pereira et al in Human Mol Genetics 2011 or Pereira et al Cell cycle 2012, or another article on this topic in Frontiers in Physiology 2014 by Fetahu IS et al.

3.       Please, clearly indicate that not only the kidney produces the hormonal form of vitamin D. Immune cells were the first evidence of local capacity to convert 25(OH)D into 1,25-dihydroxycholecalciferol. Furthermore, monocyte-macrophages also express 25-hydroxylases,   so that they can also fully activate vitamin D (cholecalciferol) into 25-hydroxyvitamin D and the final active hormone 1,25-dihydroxyvitamin D.

4.       Lane 87. 1,25-dihydroxyvitamin D is a potent steroid hormone. This is not an over statement.

It acts as a steroid hormone through binding to its receptor to transactivate or transrepress more than 900 genes, many of them, as extensively described fro key immune disorters , involved in immune regulation.

5.       Please, emphasize the marked differences in the amount of vitamin D supplementation required for different auto immune disorders, such as the high levels required for MS vs. the normalization of circulating vitamin D sufficient to prevent the onset of tuberculosis.

Author Response

Thank you very much for your enriching comments. 
In topics 1., 3. and 4. - all the necessary substantive changes were made to the indicated fragments. We would like to thank you for your careful analysis of the text. The changes implemented are certainly an important element in the efforts to refine the manuscript.
Topic 2. The section on epigenetics was expanded based on the suggested sources, devoting much attention to the process of demethylation and the influence of vitamin D on this process. At the same time, other epigenetic topics were also enriched with current reports.
Topic 5. Specific data on recommended supplementation doses and the differences between the recommendations of various scientific bodies and between the purpose of application were added. Attention was given to the unique supplementation in multiple sclerosis while highlighting the phenomenon of vitamin D toxicity in the use of extremely high doses.

We believe that all the modifications suggested by the reviewers and implemented by us will ensure that the article will become more interesting and clear, and will receive positive feedback.